# DGNet: Self-Supervised Delta2Gamma Multi-Band EEG Representation Learning for Dementia Classification

## Abstract

As the global population ages and dementia cases rise, there is an urgent need for effective early diagnosis and monitoring of neurodegenerative diseases. Electroencephalogram (EEG)-based technologies are increasingly important due to their portability, affordability, and suitability for widespread screening compared to other neuroimaging methods. However, EEG signals present challenges such as low signal-to-noise ratio, high inter-subject variability, and limited labeled data, especially in elderly or dementia patients, which restricts the effectiveness of traditional supervised learning approaches. Leveraging the neurophysiological significance of the five EEG frequency bands (delta, theta, alpha, beta, gamma), this study introduces an innovative multi-head Simple Framework for Contrastive Learning of Visual Representations (SimCLR) architecture. The proposed Delta2Gamma (DGNet) model combines frequency-band specific representation learning, enabling more precise detection of subtle EEG changes linked to brain disorders like dementia. Our self-supervised learning (SSL) adaptive multi-band heads model achieved a 31.5% relative performance improvement over training from scratch, and a 25.4% improvement over the single-head approach. To the best of our knowledge, our proposed method achieved state-of-the-art performance in multi-head approaches. The source code is available at GitHub by `https://anonymous.4open.science/r/iclr2026-7FE2`.

## 1 Introduction

At the turn of the 21st century, humanity is facing one of the greatest challenges in the history of public health. It is a complex crisis caused by the collision of two massive trends: the global aging of the population at an unprecedented rate and its inevitable corollary (Nichols et al., 2022), the explosion in the prevalence of dementia (Li et al., 2022). These two phenomena are more than just demographic shifts or individual diseases; they are a tsunami that is shaking the very foundations of socioeconomic structures and healthcare systems around the world. A clear understanding of the scale and urgency of the problem is a prerequisite for finding innovative solutions (Olivari et al., 2020).

Traditional diagnostic methods for dementia, such as magnetic resonance imaging (MRI) and positron emission tomography (PET), have limitations in early detection due to their high costs and limited accessibility (Haidar et al., 2023; Juganavar et al., 2023). Additionally, these tests can only be performed at medical institutions operated by specialized healthcare professionals, making it difficult for the general public to regularly monitor their cognitive health in daily life (Kancharla, 2024). Early detection of mild cognitive impairment (MCI), the precursor stage of dementia, is particularly challenging (Sabbagh et al., 2020), and as a result, most patients are only diagnosed after their symptoms have significantly progressed (Cavedoni et al., 2020).

Despite the vast scale and severity of the dementia crisis, current clinical responses particularly in the diagnostic process are fundamentally inadequate to address the issue. The existing diagnostic paradigm relies on technologies that are not scalable, leading to a serious "diagnostic bottleneck" worldwide. Precisely analyzing the causes of this bottleneck is essential for setting the right course toward a solution. These technologies have critical limitations that prevent them from serving as

frontline screening tools for large populations. As explicitly noted in the research literature, MRI and PET are fundamentally constrained by extremely high costs and limited accessibility. Because they require large, stationary, and expensive equipment, as well as highly specialized facilities and personnel, their benefits are inevitably restricted to only a small fraction of the population.

As an alternative to overcoming the current diagnostic bottleneck, electroencephalography (EEG) offers a promising solution that combines scientific validity with practical advantages. EEG is a non-invasive method that can directly and quantitatively measure declines in brain function (Dabbabi et al., 2023), and it has the potential to overcome the limitations of existing technologies and establish a scalable diagnostic paradigm. This represents a fundamental distinction from functional magnetic resonance imaging (fMRI), which measures blood flow changes that serve as indirect proxies for neural activity (Yen et al., 2023). By directly capturing electrical signals generated by neurons, EEG most directly reflects how the neuropathological changes defining dementia impact brain function (Smailovic & Jelic, 2019). Consequently, EEG analysis can provide critical insights for understanding and diagnosing the underlying neurophysiological basis of cognitive decline (Yuan & Zhao, 2025).

A substantial body of prior research has identified characteristic and quantifiable "spectral signatures" associated with Alzheimer's disease (AD) and other forms of dementia, as also supported by the technical analysis provided. The core of this signature can be summarized as an overall "slowing" of brain oscillations. The specific biomarkers are as follows:

- **Increased power in low-frequency bands**: Patients with AD show a statistically significant increase in the power spectral density (PSD) of low-frequency bands, such as delta ($\delta$, 0-4 Hz) and theta ($\theta$, 4-8 Hz), compared to healthy controls (Moretti et al., 2004). Notably, the literature (Baik et al., 2022) states that profuse theta waves with age depict abnormal activity, emphasizing that excessive theta activity is indicative of pathological conditions.

- **Decreased power in high-frequency bands**: This slowing of brain activity is accompanied by a reduction in power in the higher-frequency bands (Benwell et al., 2020). This includes a decrease in alpha ($\alpha$, 8-12 Hz) waves, which are associated with relaxed wakefulness, and beta ($\beta$, 12-30 Hz) waves, which are linked to active concentration. Most notably, there is a marked reduction in gamma ($\gamma$, 30$\sim$ Hz) waves, which are essential for higher cognitive functions such as short-term memory (Kamiński et al., 2011). The literature explicitly states, gamma waves decline with cognitive deterioration, clearly indicating that changes in gamma activity are a key marker of cognitive decline (Traikapi & Konstantinou, 2021).

This research perfectly aligns with the future direction of dementia research and treatment, which aims to integrate artificial intelligence, digital health tools, and personalized medicine. The development of home-based cognitive function tests or personalized assistive technologies demonstrates a clear trend toward shifting the focus of care from hospitals to patients. The EEG system proposed in this study will serve as a foundational technology for building this future medical ecosystem. The main contributions of this paper are summarized as follows:

1. **Frequency-band specific Encoding**: We propose Delta2Gamma (DGNet) architecture that decomposes EEG signals into five standard frequency bands ($\delta$, $\theta$, $\alpha$, $\beta$, and $\gamma$) for processing, enabling the extraction of frequency-band specific representations.

2. **Multi-Band Head**: Each frequency band is processed by an independent CNN encoder and projection head, thereby preserving neural information unique to each band.

3. **Effective for dementia classification**: We evaluate a self-supervised learning model specifically tailored to the neurophysiological characteristics of EEG signals in the dementia classification task.

## 2 PROPOSED METHOD

The proposed framework is illustrated in Figure 1. The overall training process consists of two stages: pre-training and linear evaluation. We perform contrastive learning on unlabeled EEG data. During this process, the model learns the general characteristics and patterns of EEG signals, and data augmentation enables it to acquire robust feature representations against various transformations. The following sections discuss these components in detail.

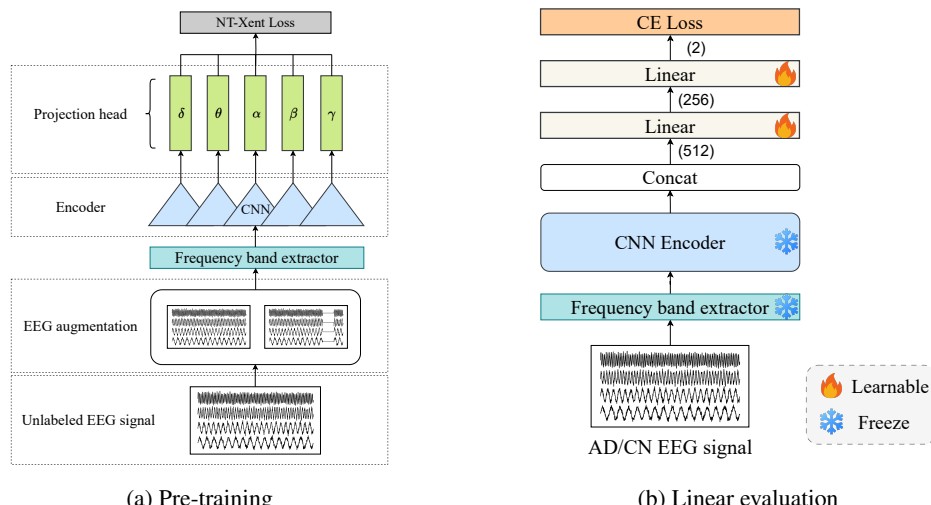

(a) Pre-training        (b) Linear evaluation

Figure 1: Overview of the entire learning process of our proposed Delta2Gamma (DGNet) model. (a) Pre-training phase based on self-supervised learning: Data augmentation is applied to the unlabeled EEG data. Then, using adaptive Normalized Temperature-scaled cross entropy (NT-Xent) contrastive loss with regularization, the encoder is trained to learn meaningful feature representations across multiple frequency bands ($\delta, \theta, \alpha, \beta, \gamma$). (b) Linear evaluation phase: A classifier is added on top of the pre-trained encoder. The pre-trained encoder is frozen during this stage. Then, the entire model is intentionally retrained using labeled data for specific tasks such as classifying Alzheimer's disease (AD) and cognitively normal (CN) groups. The numbers on the linear layers indicate the dimensionality of each layer.

## 2.1 ARCHITECTURES

**Self-Supervised learning (SSL)** Simple Framework for Contrastive Learning of Visual Representations (SimCLR) is a self-supervised learning method developed by the Google Brain team (Chen et al., 2020). It offers an innovative approach for learning useful feature representations from large amounts of unlabeled data. The core idea of this method is contrastive learning, where different augmented versions of the same sample are encouraged to be close to each other in the feature space, while samples from different instances are pushed apart.

Our implementation of SimCLR, as shown in Figure 1(a), is specifically tailored for EEG signals. This advanced adaptation applies the original image-based SimCLR framework to neural signal analysis. The model simultaneously processes multiple frequency bands of EEG signals ($\delta$, $\theta$, $\alpha$, $\beta$, $\gamma$) and employs independent projection heads for each frequency band, enabling more fine-grained feature learning. This SimCLR-based self-supervised learning approach is an innovative methodology that overcomes the limitations of traditional supervised learning. In the medical field, for example, applications such as dementia diagnosis using EEG signals often face challenges in acquiring large amounts of labeled data, whereas unlabeled EEG data can be collected relatively easily. In such scenarios, SimCLR enables the learning of rich feature representations from unlabeled data, allowing high-performing classification models to be built even with only a small amount of labeled data.

**Multi-Band** Our most innovative aspect lies in the processing of multiple frequency bands. EEG signals are composed of several frequency bands, each reflecting different brain activities and carrying unique neurological significance. We decompose the original signal into five frequency bands ($\delta$: 0.5-4 Hz, $\theta$: 4-8 Hz, $\alpha$: 8-13 Hz, $\beta$: 13-30 Hz, $\gamma$: 30-45 Hz) and perform independent feature extraction for each band.

This approach separates EEG signals into five major frequency bands ($\delta$, $\theta$, $\alpha$, $\beta$, $\gamma$) and extracts unique features from each band, as illustrated in Figure 2. The first stage involves a frequency band extractor that separates features corresponding to each frequency band from the original multi-

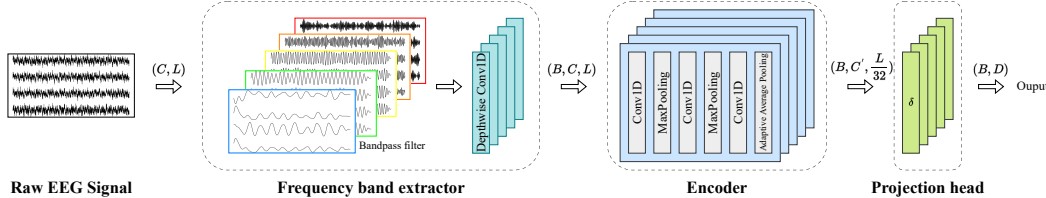

Figure 2: Process of frequency band feature representation. The input EEG signal is split into five frequency bands ($\delta, \theta, \alpha, \beta, \gamma$) using parallel 1-dimensional depthwise convolution. Each frequency-band specific feature is independently encoded, and projection heads generate a final feature vector for each band. ($B$: number of frequency bands, $C$: number of EEG channels, $C'$: number of encoded channels, $D$: the output dimension, $L$: number of time points)

channel EEG signal with shape $[C, L]$, where $C$ is the number of channels and $L$ is the signal length. In our experiment, we set the value of $C$ to 19, and $L$ was defined as the number of seconds multiplied by the sampling rate of 500 Hz. This module consists of five parallel 1-dimensional depthwise convolution layers. Each convolution layer is configured with a kernel size of 7 and padding of 3, preserving the input signal length $L$, followed by batch normalization (Ioffe & Szegedy, 2015) and Rectified Linear Units (ReLU) activation (Agarap, 2018). With groups set to $C$, each input channel is processed independently by its own 1-dimensional convolution filter, minimizing inter-channel information mixing and allowing effective learning of the unique temporal patterns of each channel (Lawhern et al., 2018).

The input to the model is a raw EEG signal with shape $[C, L]$. First, the signal is decomposed into five canonical frequency bands using bandpass filters. Each filtered signal retains the original shape, resulting in five parallel representations $[5, C, L]$. Each band is then independently processed by a 1-dimensional convolution encoder with reduced depth and downsampling operations, designed to efficiently extract temporal and spatial features while minimizing computational overhead. For each frequency band, the encoder consists of three convolutional blocks with increasing channel dimensions ($32 \rightarrow 64 \rightarrow 128$), interleaved with batch normalization, ReLU activation, and max pooling layers. To summarize temporal dynamics, a global average pooling layer is applied to each frequency-band specific representation. The output after passing through the encoder is $[5, C, L/32]$, and the pooled outputs from all five bands are then utilized. The pooled outputs from all five bands are concatenated and passed through a feature fusion comprising a fully connected layer, batch normalization, ReLU activation. This results in a compact $[5, 128\text{-dimensional}]$ embedding that captures multi-band EEG characteristics.

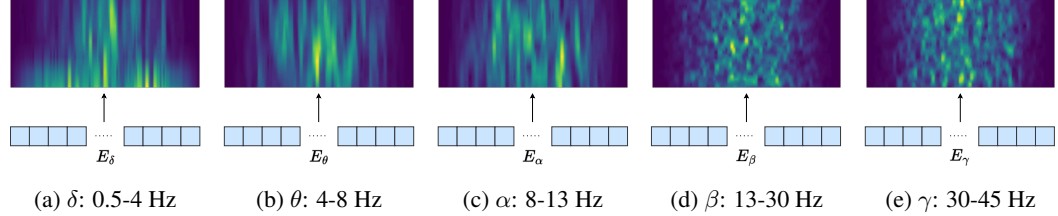

(a) $\delta$: 0.5-4 Hz      (b) $\theta$: 4-8 Hz      (c) $\alpha$: 8-13 Hz      (d) $\beta$: 13-30 Hz      (e) $\gamma$: 30-45 Hz

Figure 3: Spectrogram visualization of embeddings from the encoder, which incorporates the pretrained feature extractor, across the 5 EEG frequency bands. The x-axis represents time, while the y-axis denotes the frequency ranges corresponding to each band.

The projection head receives each of the $5\times(128\text{-dimensional})$ feature vectors output by the encoders. This design produces $5\times(128\text{-dimensional})$ feature vectors, one for each frequency band. These vectors encapsulate rich and diverse information from the multi-channel EEG signals and can be effectively used as input for further analysis or diagnostic models. The spectrogram visualization of embeddings from the encoder, which includes the pretrained feature extractor, across the five EEG bands is shown in Figure 3.

**Downstream Task**  In this study, the classification model utilizes a pre-trained encoder on multi frequency band representations. Two approaches were considered for training: In the first approach, the encoder's parameters are kept frozen, and only the newly added classifier is trained. This process is illustrated in Figure 1b. This allows us to assess the quality of the features extracted by the pre-trained encoder. In the second approach, known as linear evaluation, all parameters of the model including those of the encoder are updated during training to adapt to the specific classification task.

The classifier, which is attached to the final layer of the pre-trained encoder, is a newly constructed neural network with a MLP architecture. It takes the feature vector output by the encoder as input and predicts the final class. The classifier consists of three linear layers: the first hidden layer contains 512 nodes, and the second hidden layer contains 256 nodes. After each hidden layer, a ReLU activation function (Agarap, 2018) is applied to introduce non-linearity, and batch normalization is used to stabilize the training process. To prevent overfitting and improve the generalization performance of the model, dropout (Srivastava et al., 2014) is applied to each hidden layer at rates of 0.3 and 0.2, respectively. The final output layer produces logits with a dimensionality equal to the number of target classes.

## 2.2 DATA AUGMENTATION

The core of self-supervised learning lies in extracting meaningful signals from the data itself through pretext tasks. In line with this philosophy, we apply various transformations to the original EEG signals to generate two views that are semantically identical but morphologically different. It generates two views of the same original signal that are semantically identical but morphologically different by applying various transformations. Specifically, techniques such as Gaussian noise addition, amplitude scaling, time domain masking, frequency domain masking, and channel dropout are used to create diverse perspectives from the same EEG signal, as illustrated in Figure 4. For each case, we apply the following augmentations: Gaussian noise with a standard deviation of 0.03, amplitude scaling by a random factor between 0.8 and 1.2, and random masking of 10% in both the time and frequency domains. Additionally, with a probability of 10%, we perform channel dropout on 10% of the total channels.

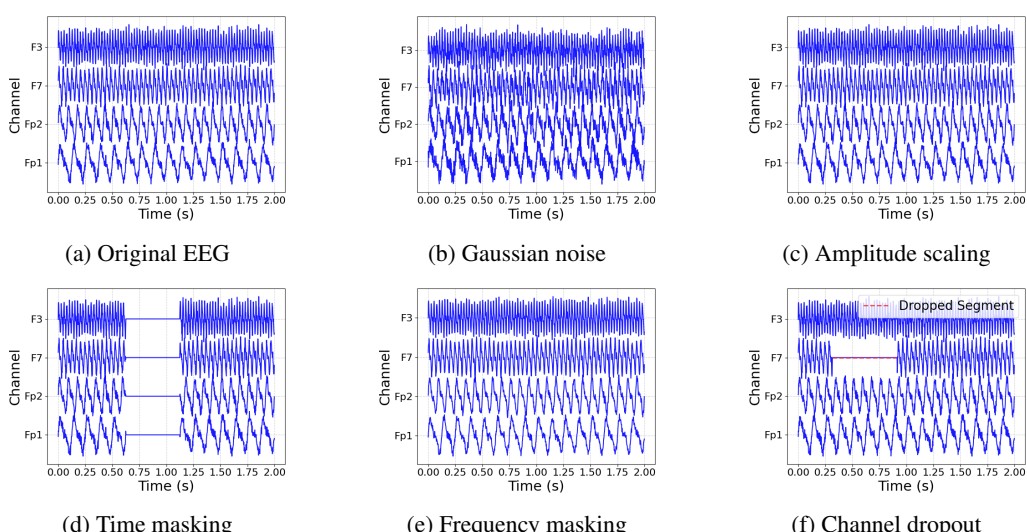

Figure 4: Examples of studied EEG data augmentation methods are shown: (a) original EEG, (b) Gaussian noise addition, (c) amplitude scaling, (d) time domain masking, (e) frequency domain masking, and (f) channel dropout. Each augmentation can probabilistically transform the data using certain internal parameters, such as scale range or noise level.

These augmented views are then used in an instance discrimination task. In this setup, two augmented views derived from the same original signal should have similar representations in the feature space (positive pair), while views derived from different original signals should have distinct representations (negative pairs). Through this contrastive learning process, the model learns to cap-

ture the essential characteristics of the signal and acquire feature representations that are robust to noise and transformations.

## 2.3 TRAINING OBJECTIVE

This section describes losses for training, including, adaptive Normalized Temperature-scaled cross entropy (NT-Xent) and regularization (Wang et al., 2024). The final loss is defined as $\ell = \sum_{b=1}^{B} \ell_b$, where $\ell_b$ denotes the loss of the $b$-th frequency band head as defined, $B$ refers to the number of frequency band heads.

$$\ell_i = \sum_{b=1}^{B} \left( -\frac{1}{\tau_i^{(b)+}} \operatorname{sim}(\mathbf{z}_i^{(b)}, \mathbf{z}_i^{(b)+}) + \frac{1}{\tau_{(i,n^*)}^{(b)-}} \max_{n=1,\cdots,N} \operatorname{sim}(\mathbf{z}_i^{(b)}, \mathbf{z}_{(i,n)}^{(b)-}) + \beta\,\Omega(\tau_i^{(b)+}) - \beta\,\Omega(\tau_{(i,n^*)}^{(b)-}) \right) \tag{1}$$

In the above equation, the function $\operatorname{sim}(\cdot,\cdot)$ denotes cosine similarity, which can correspond to either a positive pair $(\mathbf{z}_i^{(b)}, \mathbf{z}_i^{(b)+})$ or a negative pair $(\mathbf{z}_i^{(b)}, \mathbf{z}_{(i,n)}^{(b)-})$, $n$ denotes the individual indices of negative samples $(1, 2, ..., N)$, and $N$ is the total number of negative samples for the $i$-th anchor. The $\tau_i^{(b)+}$ and $\tau_{(i,n)}^{(b)-}$ represent learnable adaptive positive and negative temperatures, respectively; the asterisk "$*$" in $\tau_{(i,n^*)}^{b-}$ indicates the index $n^* = \arg\max_{n=1,\cdots,N} \operatorname{sim}(\mathbf{z}_i^{(b)}, \mathbf{z}_{(i,n)}^{(b)-})$. $\beta \geq 0$ controls the temperature regularization imposed by $\Omega(\cdot)$. We describe the $\Omega(\cdot)$ function in more detail in section Regularization.

**NT-Xent**  Normalized Temperature-scaled cross entropy loss function, which is the core of Sim-CLR, provides a mathematical basis for contrastive learning. In the attached code, the multi-head implementation computes independent NT-Xent losses for each frequency band and combines them through a weighted average to obtain the final loss. Additionally, an adaptive temperature mechanism introduces a normalization term that accounts for the learning difficulty of each band, enabling more stable and effective training.

$$\ell_i = -\log \frac{\exp\left(\operatorname{sim}(\mathbf{z}_i, \mathbf{z}_i^+)/\tau\right)}{\sum_{n=1}^{N} \exp\left(\operatorname{sim}(\mathbf{z}_i, \mathbf{z}_{(i,n)}^-)/\tau\right)} \tag{2}$$

This sophisticated loss function aims to maximize the similarity between positive pairs while maximizing the distinction between negative pairs. Through this process, the model learns the intrinsic structure and patterns of the data. In the case of EEG signals, since each frequency band carries distinct neurological significance, performing independent contrastive learning for each band allows the model to capture more fine-grained and meaningful.

**Regularization**  Furthermore, we implement an advanced contrastive learning loss function that applies adaptive temperature to each frequency band head. While traditional SimCLR uses a fixed temperature parameter, this implementation dynamically adjusts the temperature based on the characteristics of each frequency band head. This approach considers the unique distribution properties and learning difficulty of different frequency bands, enabling more effective contrastive learning. The regularization is formulated as follows:

$$\Omega(\tau) = (d'/2)\log(\tau) + 1/\tau, \tag{3}$$

where $d$ is the feature dimension. The projection band head maps these $d$-dimensional vectors into $d'$-dimensional vectors. This induces the temperature $\tau$ to move to $\tau = 2/d'$.

## 3 EXPERIMENTAL SETUP

During the pre-training stage, the model was trained using the AdamW optimizer (Loshchilov & Hutter, 2017) with a batch size of 64 and a learning rate of $1 \times 10^{-4}$. The adaptive NT-XENT loss function was employed, with the adaptive temperature parameter adjusted between 0.05 and 0.5, and a temperature regularization parameter $\beta$ of 0.01 applied. In the subsequent linear evaluation stage, Leave-One-Subject-Out (LOSO) cross-validation was used, and classification was performed with the pre-trained encoder weights kept frozen. The AdamW optimizer was used with a batch size of 32 and a learning rate of $1 \times 10^{-4}$ to minimize the cross-entropy loss. Both stages were trained for up to 100 epochs, with early stopping applied if no performance improvement was observed for 10 consecutive epochs. We trained the model on Intel Xeon-Silver 4410Y CPU and NVIDIA GeForce RTX 4090 GPU. In addition, a weight decay (Loshchilov & Hutter, 2017) of $1 \times 10^{-5}$ was applied. The scheduler uses cosine annealing with warm restarts (Loshchilov & Hutter, 2016).

### 3.1 DATASET

The dataset (Miltiadous et al., 2023b) used in this study consists of resting-state, eyes-closed EEG recordings from a total of 88 participants. Among them, 36 were diagnosed with Alzheimer's disease (AD) group, 23 with frontotemporal dementia (FTD) group, and 29 were cognitively normal (CN) group. Cognitive and neuropsychological status was assessed using the mini-mental state examination (MMSE), with scores ranging from 0 to 30, where lower scores indicate greater cognitive impairment. The median disease duration was 25 months, and no dementia-related comorbidities were reported in the AD group. The mean MMSE scores were 17.75 for the AD group, 22.17 for the FTD group, and 30 for the CN group. The mean ages were 66.4 years for the AD group, 63.6 years for the FTD group, and 67.9 years for the CN group.

EEG recordings were collected by a team of experienced neurologists at the department of neurology, AHEPA University Hospital of Thessaloniki, using the Nihon Kohden EEG-2100 clinical system. 19 scalp electrodes (according to the 10-20 system) and two mastoid reference electrodes were used, with impedance maintained below 5 k. All recordings were performed with participants seated and eyes closed, at a sampling rate of 500 Hz and a sensitivity of 10 $\mu$V/mm. Both bipolar and referential montages (referenced to Cz) were included. The average recording duration was approximately 13.5 minutes for the AD group (5.1-21.3 min), 12 minutes for the FTD group (7.9-16.9 min), and 13.8 minutes for the CN group (12.5-16.5 min), resulting in a total of 485.5, 276.5, and 402 minutes of data for each group, respectively.

### 3.2 PREPROCESSING

First, when performing EEG source localization, it is recommended to calculate the average reference. Since referencing affects the measurement of signal amplitudes, it is an important step in EEG preprocessing. If a single electrode is used as a reference for the others, brain activity and noise from the reference electrode can be mixed into the signals. One approach is to compute the average potential across all channels and use this average as the reference. This way, the potential at each scalp location is compared to all other recorded sites, better reflecting the unique characteristics of each location. EEG data were preprocessed by applying a 6th order Butterworth bandpass filter in the 0.5 to 45 Hz range (Nour et al., 2024), which effectively preserves information relevant to distinguishing neural activity between AD and CN. Additionally, artifacts were removed using blind source separation with independent component analysis (ICA) (Comon, 1994; Makeig et al., 1995; Jung et al., 1997). All of the processing for the EEG was done with the python library MNE (Gramfort et al., 2013).

### 3.3 SEGMENTATION

In recent AD research, the analysis of EEG characteristics related to sleep has emerged as an important trend. In particular, EEG data collected during the eyes-close state are widely used in early diagnosis studies of dementia, as this approach minimizes the influence of external stimuli and more clearly reflects intrinsic brain activity. In line with this trend, our study segmented EEG data acquired during the eyes-close state into 30-second intervals for analysis (Ye et al., 2023; Herzog et al., 2023; Park et al., 2025). This segmentation matches the standard epoch length used in sleep re-

search, thereby enhancing the reliability and comparability of our investigation into the relationship between dementia and sleep.

### 3.4 Leave-One-Subject-Out

In EEG research, the LOSO method (Kunjan et al., 2021) is a widely used cross-validation technique for rigorously evaluating the generalization performance of a dataset. In this approach, one subject is used as the validation set while all remaining participants are used as the training set. This process is repeated for each participant in the dataset. In other words, if the dataset consists of $N$ subjects, the procedure is repeated $N$ times, with each iteration using the data from one subject for testing and the data from the remaining $(N - 1)$ subjects for training.

The main purpose of this method is to account for the high inter-individual variability in EEG signals and to assess how well the model can predict data from new, unseen subjects. Because EEG data can vary greatly due to individual physiological characteristics, scalp conditions, noise, and other factors, a more stringent and realistic performance evaluation is required compared to simple k-fold cross-validation. The LOSO method addresses this by preventing data leakage between subjects and ensuring complete independence between the training and validation sets.

## 4 Results and Discussion

### 4.1 Comparison of Classification Performance

To evaluate the performance of the proposed model, we compared it with major benchmark models in the field of EEG analysis. The details of each EEG benchmark model are provided in the appendix, and for the SSL models, fine-tuning was performed when pretrained weights were available. As shown in Table 1, the proposed model achieved 92.90% accuracy and 92.85% F1-score, significantly outperforming all comparison models. These results clearly demonstrate the superiority of our approach, indicating that the proposed adaptive multi-band head method is highly effective in learning and classifying the complex features of EEG signals.

| Model | Backbone | Acc (%) | F1 (%) |
|---|---|---|---|
| ATCNet (Altaheri et al., 2022) | CNN & RNN & Att | 74 | 74 |
| BIOT (Yang et al., 2023) | Att | 53 | 40 |
| CTNet (Zhao et al., 2024) | CNN & Att | 74 | 73 |
| Deep4Net (Schirrmeister et al., 2017) | CNN | 49 | 49 |
| EEGConformer (Song et al., 2022) | CNN & Att | 57 | 54 |
| EEGInception (Santamaria-Vazquez et al., 2020) | CNN | 39 | 37 |
| EEGNet (Lawhern et al., 2018) | CNN | 46 | 45 |
| FBCNet (Mane et al., 2021) | CNN | 48 | 38 |
| Labram (Jiang et al., 2024) | CNN & Att | 54 | 38 |
| S-JEPA (Guetschel et al., 2024) | CNN & Att | 50 | 50 |
| SPARCNet (Jing et al., 2023) | CNN | 54 | 53 |
| TIDNet (Kostas & Rudzicz, 2020) | CNN | 44 | 40 |
| **Ours (adaptive 5 band heads)** | CNN | **93%** | **93%** |

Table 1: Comparison of the Alzheimer's disease (AD) and the cognitively normal (CN) classification performance between the proposed adaptive 5 band heads model and leading benchmark models in EEG analysis, including both supervised and self-supervised learning approaches. Here, CNN refers to convolutional neural networks, RNN to recurrent neural networks, and Att to attention mechanisms.

### 4.2 LOSO Performance

To further validate the performance of the proposed model, we compared its results with those of previous studies on the task of classifying AD and CN subjects. For fair evaluation and to assess generalization performance, all models were evaluated using strict LOSO cross-validation. As shown

in Table 2, the proposed adaptive 5 band heads model achieved 92.90% accuracy and an 92.85% F1-score. These results demonstrate that the proposed multi-frequency band-based self-supervised learning and linear evaluation approach is highly effective in overcoming inter-subject variability and learning features with excellent generalization performance.

| Model | Acc (%) | F1 (%) | Pre (%) | Rec (%) |
|---|---|---|---|---|
| kNN (Ntetska et al., 2025) | 60.30 | 58.90 | 57.90 | 59.90 |
| CNN (Stefanou et al., 2025) | 79.45 | 77.60 | 76.32 | 76.06 |
| Random Forest (Sarkar et al., 2025) | 80.00 | 81.69 | - | 80.55 |
| DICE-Net (Miltiadous et al., 2023a) | 83.28 | 84.12 | 88.94 | 79.81 |
| CNN (Vo et al., 2025) | 84.62 | 86.11 | - | 86.11 |
| MJANet (Sun et al., 2025) | 85.23 | 86.37 | 88.12 | 84.69 |
| Dual-Branch (Chen et al., 2023) | 85.78 | - | - | 83.22 |
| Random Forest (Parihar & Swami, 2024) | 88.90 | - | - | - |
| BI-MCGNN (Zhang & Zhu, 2025) | $91.25 \pm 0.38$ | - | - | $\mathbf{93.32 \pm 0.46}$ |
| **Ours (adaptive 5 band heads)** | **92.90** | **92.85** | **93.27** | 92.90 |

Table 2: Leave-One-Subject-Out (LOSO) performance comparison for Alzheimer's disease (AD) and the cognitively normal (CN) classification using the dataset (Miltiadous et al., 2023b).

## 4.3 ABLATION STUDY

We analyze the impact of each component of the proposed model on performance, as summarized in Table 3. To compare with self-supervised learning, we trained the CNN model from scratch on the EEG dataset, and it achieved an accuracy of 63.35%. The experimental results show that using only a single projection head (single-head) led to a significant drop in accuracy to 73.52%, , while the 5 multi-head architecture achieved 79.55%, highlighting the importance of the multi-frequency band head architecture. Without data augmentation, we masked 15% of the EEG signal and trained the encoder model to reconstruct it using mean squared error (MSE) loss, achieving 78.58% accuracy. In addition, fixing the temperature ($\tau$) parameter at 0.1 and removing regularization resulted in decreased performance, with accuracies of 86.53% and 90.64%, respectively. These findings suggest that both the adaptive temperature adjustment and the regularization techniques in the proposed model contribute positively to the final performance.

| Model | Acc (%) | F1 (%) | Pre (%) | Rec (%) | AUC (%) |
|---|---|---|---|---|---|
| **Adaptive 5 band heads** | **92.90** | **92.85** | **93.27** | **92.90** | **96.77** |
| w/o self-supervised learning | 63.35 | 61.68 | 63.77 | 63.35 | 67.98 |
| Single-head | 73.52 | 72.33 | 82.88 | 73.52 | 64.89 |
| w/o augmentation | 78.58 | 78.05 | 79.74 | 78.58 | 78.34 |
| Multi-head (5 heads) | 79.55 | 79.38 | 82.77 | 79.55 | 78.04 |
| constant temperature ($\tau = 0.1$) | 86.53 | 86.56 | 87.30 | 86.53 | 87.21 |
| w/o regularization | 90.64 | 90.59 | 91.72 | 89.64 | 91.33 |

Table 3: Ablation Study results for the proposed model on the dataset (Miltiadous et al., 2023b).

## 5 CONCLUSION

We propose a multi-head SimCLR-based framework for contrastive learning of EEG representations, leveraging independent CNN encoders and adaptive temperature parameters for each of the 5 frequency bands. Using Adaptive Multi-head Contrastive Learning (AMCL) strategy (Wang et al., 2024), we compute and aggregate contrastive losses for each band, resulting in superior representation learning and classification performance, especially with limited labels.

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

# A BASELINES

Here, we present the details of the baseline models compared with DGNet. The model summary is shown in Table 4.

**ATCNet** (Altaheri et al., 2022) is a convolution-first architecture enhanced with a lightweight Attention-TCN sequence module. The end-to-end flow is as follows: (i) it learns temporal filter banks and spatial projections (in the style of EEGNet), transforming downsampled time into a compact feature map, (ii) a sliding window extracts overlapping temporal segments from this map, (iii) for each segment, a small multi-head self-attention over time is applied, followed by causal and dilated convolutions, (iv) the segment-level features are then aggregated (either by averaging window logits or concatenation) and mapped through a linear layer with a max-norm constraint.

**BIOT** (Yang et al., 2023) is a large language model for biosignal classification. It serves as a wrapper around the BIOTEncoder and ClassificationHead modules. It is designed for N-dimensional biosignal data such as EEG and electrocardiogram (ECG).

**CTNet** (Zhao et al., 2024) is an integrated neural network designed to classify motor imagery (MI) tasks using EEG signals. By combining convolutional neural networks with transformer encoders, it effectively models both short-range and long-range temporal patterns in the EEG data.

**Deep4Net** (Schirrmeister et al., 2017) is a deep learning convolutional neural network architecture designed for EEG signal decoding and classification. It uses multiple convolutional layers to automatically extract hierarchical features from raw EEG data, supporting efficient and accurate classification in EEG-based brain-computer interfaces or neurological condition diagnosis.

**EEGConformer** (Song et al., 2022) is a convolution-first model equipped with a lightweight transformer encoder. The end-to-end flow is as follows: (i) continuous EEG signals are transformed into a compact sequence of tokens through a temporal-spatial convolutional stem followed by temporal pooling, (ii) a small multi-head self-attention mechanism is applied across the tokens to integrate longer-range temporal context, (iii) the token sequence is aggregated and fed to a linear readout layer, adding sufficient attention capacity to capture dependencies beyond the pooling scope while preserving the strong inductive bias of the shallow CNN filter bank.

**EEGInception** (Santamaria-Vazquez et al., 2020) is based on the original InceptionNet for computer vision. Its main goal is to extract features at multiple scales in parallel. The network consists of two blocks, each composed of three Inception modules with skip connections.

**EEGNet** (Lawhern et al., 2018) is a compact convolutional network designed for EEG decoding with a pipeline that reflects traditional EEG processing. It (i) learns temporal frequency-selective filters, (ii) learns spatial filters for those frequencies, and (iii) compresses the features with depthwise separable convolutions before a lightweight classifier.

**FBCNet** (Mane et al., 2021) is inspired by the FBCSP (Filter Bank Common Spatial Pattern) algorithm and applies spatial convolutions and variance calculations along the temporal axis.

**Labram** (Jiang et al., 2024) is a large brain model for learning generic representations from tremendous EEG data in BCI. It learns generic EEG representations by predicting masked neural tokens of EEG patches based on full-attention transformers. The model can be used in two modes: (i) Neural Tokenizer, designed to obtain embedding layers (e.g., for classification), and (ii) Neural Decoder, which extracts amplitude and phase outputs using Vector Quantized Single Neural Phase (VQSNP).

**S-JEPA** (Guetschel et al., 2024) is a self-supervised learning model designed for EEG signal representation. It uses a unique spatial block masking strategy tailored to EEG channel arrangements, combined with a CNN-based local encoder that encodes each EEG channel independently into embeddings. The model leverages joint embedding predictive architectures (JEPA) to learn meaningful EEG features for downstream classification tasks across different brain-computer interface (BCI) paradigms, such as motor imagery, ERP, and SSVEP.

**SPARCNet** (Jing et al., 2023) is a temporal CNN-based deep learning model designed for EEG analysis, particularly for detecting and classifying seizure and rhythmic brain activity patterns. It focuses on capturing temporal features in EEG signals using convolutional layers, and aims to support clinical diagnosis by distinguishing various pathological EEG patterns.

**TIDNet** (Kostas & Rudzicz, 2020) is a convolutional neural network designed for EEG signal classification. It first applies temporal filtering to extract time-based features from raw EEG inputs, then uses dense spatial filtering to learn spatial relationships across EEG channels.

| Model | Application | Train | #Param |
|---|---|---|---|
| ATCNet (Altaheri et al., 2022) | General | Supervised | 113,732 |
| BIOT (Yang et al., 2023) | Sleep Staging Epilepsy | SSL | 3,183,879 |
| CTNet (Zhao et al., 2024) | Motor Imagery | Supervised | 26,900 |
| Deep4Net (Schirrmeister et al., 2017) | General | Supervised | 282,879 |
| EEGConformer (Song et al., 2022) | General | Supervised | 789,572 |
| EEGInception (Santamaria-Vazquez et al., 2020) | Motor Imagery ERP & SSVEP | Supervised | 558,028 |
| EEGNet (Lawhern et al., 2018) | General | Supervised | 2,484 |
| FBCNet (Mane et al., 2021) | Motor Imagery | Supervised | 11,812 |
| Labram (Jiang et al., 2024) | General | SSL | 5,866,180 |
| S-JEPA (Guetschel et al., 2024) | Motor Imagery ERP & SSVEP | SSL | 3,456,882 |
| SPARCNet (Jing et al., 2023) | Epilepsy | Supervised | 1,141,921 |
| TIDNet (Kostas & Rudzicz, 2020) | General | Supervised | 240,404 |
| Ours (DGNet) | Dementia | SSL | 976,635 |

Table 4: The summarization of leading benchmark models in EEG analysis. "Application" refers to commonly used application domains (e.g., motor imagery, epilepsy, sleep stages). "General" indicates that the model can be applied to multiple domains or is not focused on a specific application. "#Param" represents the calculated number of essential hyperparameters required to instantiate the model class. Here, supervised learning is referred to as "Supervised", and self-supervised learning is referred to as "SSL".

# B   DETAILED EXPERIMENTAL SETTINGS

| Hyperparameters | Settings |
| --- | --- |
| Epochs | 100 |
| Batch size | 64 |
| Learning rate | 1e-4 |
| Optimizer | AdamW |
| Weight decay | 1e-5 |
| Scheduler | CosineAnnealingWarmRestarts |
| $T_0$ | 10 |
| $T_{\mathrm{mult}}$ | 2 |
| $\eta_{\min}$ | 1e-6 |
| Temperature | [0.05, 0.5] |
| Regularization weight | 0.01 |
| Projection heads | 5 |

Table 5: Hyperparameters for Delta2Gamma (DGNet) EEG pre-training.

| Hyperparameters | Settings |
| --- | --- |
| Epochs | 100 |
| Batch size | 32 |
| Learning rate | 1e-4 |
| Optimizer | AdamW |
| Weight decay | 1e-5 |
| Scheduler | CosineAnnealingWarmRestarts |
| $T_0$ | 10 |
| $T_{\mathrm{mult}}$ | 2 |
| $\eta_{\min}$ | 1e-6 |

Table 6: Hyperparameters for downstream linear evaluation.

## C METRICS

In this section, we introduce the details of the metrics used in the paper.

**Accuracy (Acc)** refers to the proportion of samples that a model correctly predicts out of the entire dataset. In other words, it represents the ratio of cases where the predicted value matches the actual value, providing an overall measure of how often the model's predictions are correct.

$$\text{Accuracy} = \frac{TP + TN}{TP + TN + FP + FN} \quad (4)$$

**Precision (Pre)** indicates the proportion of predictions labeled as positive by the model that are actually positive. It shows how reliable the model's positive predictions are.

$$\text{Precision} = \frac{TP}{TP + FP} \quad (5)$$

**Recall (Rec)** is the proportion of actual positive samples that the model correctly predicts as positive. It measures how well the model identifies true positive cases.

$$\text{Recall} = \frac{TP}{TP + FN} \quad (6)$$

**F1** is the harmonic mean of precision and recall, providing a measure of the balance between the two. In binary classification, the weighted F1-score is calculated as the sample-size-weighted average of the F1-scores for each class, offering a fairer assessment of overall performance when class imbalance is present.

$$\text{F1} = 2 \cdot \frac{\text{Precision} \cdot \text{Recall}}{\text{Precision} + \text{Recall}} \quad (7)$$

**AUC (Area Under the ROC Curve)** is a commonly used statistic calculated as the area under the receiver operating characteristic (ROC) curve. It evaluates a model's classification performance across various thresholds, indicating how well it distinguishes between positive and negative classes. A value closer to 1 represents better classification performance.

## D SUPPLEMENTARY RESULTS

The supplementary results on the key components of the proposed methodology are presented in Figure 5. As shown in Figure 5a, the 5 band heads approach consistently maintained higher accuracy than other models, while exhibiting relatively low sensitivity to changes in the temperature ($\tau$) parameter of contrastive learning. Additionally, Figure 5b illustrates the performance variations according to the epoch length, which refers to the segmentation unit of EEG signals. As the epoch length increased from 5 seconds to 30 seconds, overall performance improved. Notably, our adaptive 5 band heads model demonstrated the most stable and superior performance across all intervals, thereby validating the effectiveness and robustness of the proposed methodology.

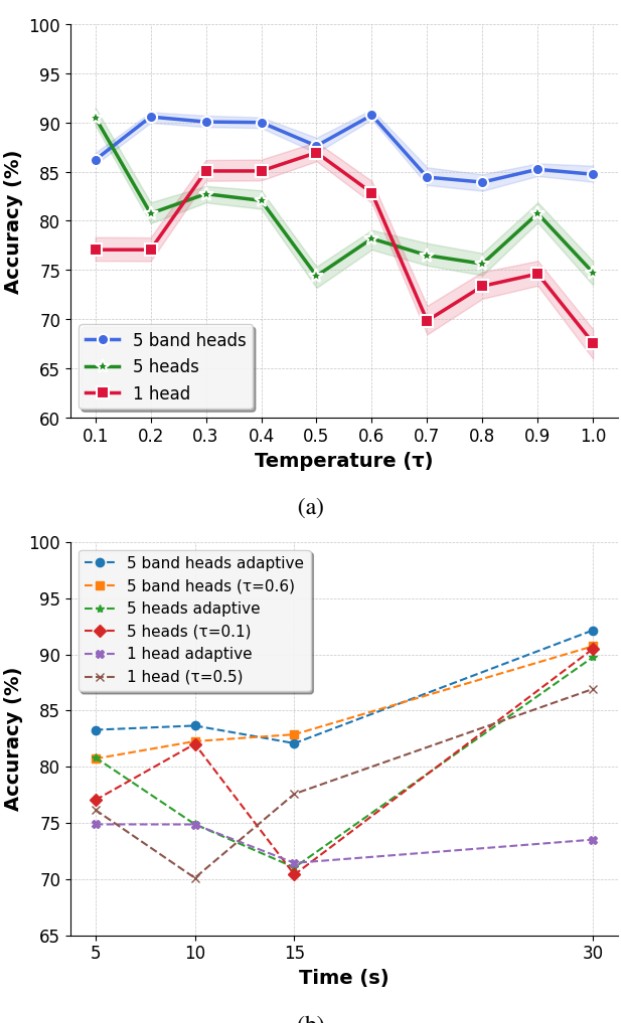

(a)

(b)

Figure 5: Supplementary results of the main components of the proposed model on the dataset (Miltiadous et al., 2023b). (a) shows the model-specific accuracy as the temperature parameter ($\tau$) of the control training varies, and (b) shows the change in accuracy as the segmentation length (s) of the input EEG signal varies.

The comparison between models with frequency band heads, multi-head, and single-head mechanisms, as shown in Figure 6, reveals a clear advantage for the 5 frequency-band heads. Across all metrics—accuracy, F1-score, precision, recall, and AUC—the model with five band heads consistently outperformed both the 5 multi-head and the single-head models. This suggests that the integration of frequency-band specific heads enhances the model's ability to capture relevant features more effectively. Furthermore, the performance gains were notable not only in accuracy but also in balanced metrics like F1-score and precision, indicating robust improvements in both detection

and classification. Overall, the frequency band head approach demonstrated superior discriminative power and reliability compared to the conventional multi-head or single-head configurations. This reinforces the benefit of specialized band heads tuned to frequency bands for this particular task.

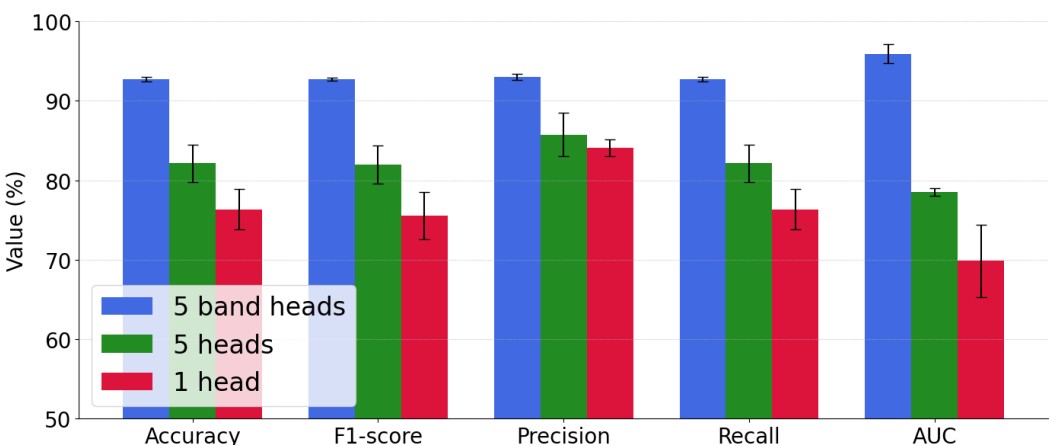

Figure 6: Performance comparison of models with frequency band heads, multiple heads, and a single head across various evaluation metrics (mean±std) on the dataset (Miltiadous et al., 2023b).

## E TOPOGRAPHY VISUALIZATION

Compared to healthy individuals, patients with dementia exhibit an increased frequency of slow brain waves ($\delta$ and $\theta$) and a characteristic decrease in $\alpha$ waves, which can be used to distinguish between the two groups. To illustrate this difference, Figure 7 shows the topography of delta band activity in EEG recordings from patients with AD and CN. The AD's group is characterized by a localized increase in delta wave activity in the frontal, temporal, and parietal lobes, which is associated with cognitive decline. In contrast, the healthy group shows relatively weak delta wave activity or a tendency for it to be distributed in the posterior regions of the brain. These spatial differences in delta wave distribution can serve as a useful biomarker for diagnosing AD and tracking its progression.

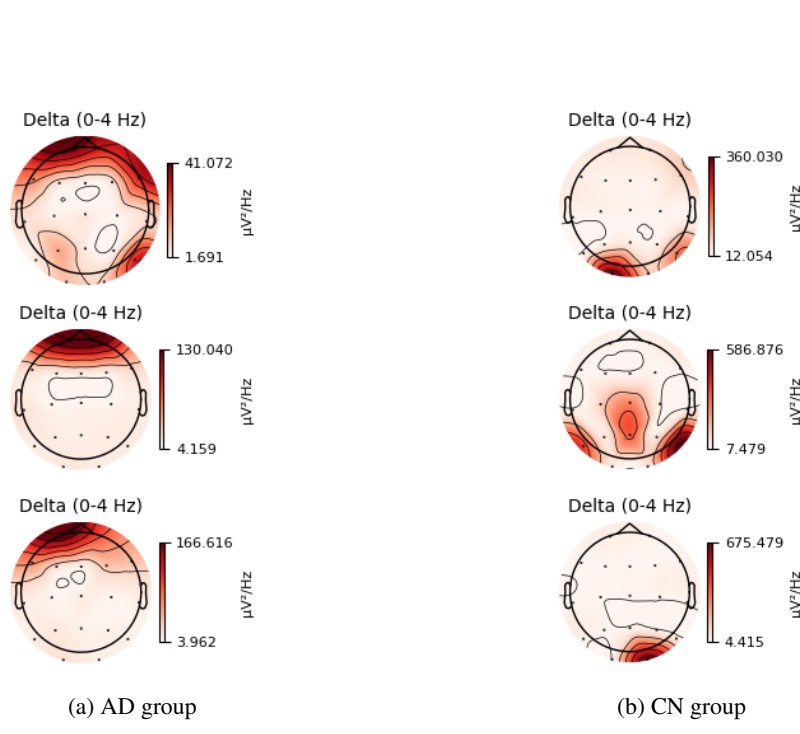

(a) AD group                                        (b) CN group

Figure 7: Distribution of delta ($\delta$) wave (0-4 Hz) brain activity in the Alzheimer's disease (AD) group (a) and the cognitively normal (CN) group (b). The AD group shows stronger activity locally in the frontal and temporal lobes, while the CN group shows relatively weaker activity in the posterior part of the brain, showing a clear difference.

