# OpenReview forum: "DGNet: Self-Supervised Delta2Gamma Multi-Band EEG Representation Learning for Dementia Classification"
_ICLR.cc/2026/Conference — ICLR 2026 Conference Withdrawn Submission_

### Official Review · Reviewer_s4Bh · 2025-10-29

**Soundness:** 1
**Presentation:** 1
**Contribution:** 1
**Rating:** 0
**Confidence:** 4

**Summary:**

DGNet introduces a self-supervised, multi-band SimCLR approach for EEG-based dementia classification, where separate encoders learn frequency-specific representations with adaptive contrastive loss. While the approach seems to be conceptually sound and tailored to EEG physiology, the evaluation on only one small dataset and the extremely high reported accuracy raise concerns about generalization and methodological robustness.

**Strengths:**

- Proposes a domain-informed self-supervised framework that integrates frequency-band–specific encoders and adaptive temperature contrastive loss, creatively adapting SimCLR to EEG data.
- Targets with EEG-based dementia classification an important and clinically relevant problem

**Weaknesses:**

- The paper reports identical results (93% accuracy, 93% F1) in Tables 1 and 2 despite claiming different validation strategies (standard classification (not clear what has been done here) vs. LOSO cross-validation). This inconsistency raises serious concerns about methodological correctness and potential data leakage. The authors should clearly describe and verify their evaluation protocol.
- The reported gains (over 20-30% improvement on a dataset of only 88 subjects) appear implausibly large compared to prior EEG-based dementia studies. Statistical significance testing, multiple-dataset validation, or external replication would strengthen credibility.
- While the adaptive multi-band SimCLR design is well-motivated, it represents an incremental adaptation of existing self-supervised learning methods rather than a fundamental methodological advance.
- The paper devotes excessive space to dementia background and EEG basics while providing insufficient technical detail on e.g., pretraining splits. In addition, it tends to overstate the role of EEG in dementia classification, presenting it as a definitive diagnostic tool rather than a complementary modality with ongoing methodological challenges.

**Questions:**

- Could you explicitly clarify how each evaluation was conducted? Were both tables based on LOSO cross-validation, or was a different split used in Table 1? How did you ensure no data leakage between pretraining and evaluation subjects, given the small dataset?
- Given the small dataset size, what measures were taken to prevent overfitting?
- Was the self-supervised pretraining performed on all subjects, including those used in the LOSO test folds?
- Please evaluate your method on additional (external) data sets.
- Were all baseline models re-implemented and trained under identical conditions, or were results taken from prior literature?
- How were hyperparameters tuned for each baseline? If tuned on the same dataset, how was data leakage prevented?
- Could you further justify why independent encoders per frequency band outperform joint processing approaches?

---

### Official Review · Reviewer_2BYL · 2025-10-31

**Soundness:** 1
**Presentation:** 1
**Contribution:** 1
**Rating:** 0
**Confidence:** 4

**Summary:**

The paper proposes DGNet, a self-superved learning model based on SimCLR for classifying dementia from EEG signals. The core architectural proposal consists of a "Multi-Band Head" approach, where the EEG signal is decomposed into five standard frequency bands (delta, theta, alpha, beta, gamma). Each band is then processed by an independent CNN encoder and projection head before the resulting features are fused for the downstream classification task

**Strengths:**

- The goal of progressing EEG for disease classification such as dementia is highly important given its severity and high prevalence.

**Weaknesses:**

- **Significant lack of novelty**: The paper's core components are standard practice and it fails to position itself within the existing literature. Analyzing the five standard EEG frequency bands is a fundamental preprocessing step in a vast majority of EEG-based studies, not a novel contribution specific to dementia. The application of SimCLR-style contrastive learning with data augmentations to EEG signals has been explored previously (e.g., [1,2,3]), but the authors do not cite or compare against any of this work.

- **Poorly motivated architecture**: The central architectural "contribution" is a simple late-fusion strategy. The paper provides no neurophysiological or machine learning justification for why processing each frequency band with a separate, independent CNN encoder is superior to standard, more efficient models that process all bands jointly (i.e., early or mid-fusion). This choice seems arbitrary and is not supported by analysis.

- **Insufficient and confusing experimental validation**: The evaluation is too weak to support the paper's claims. The entire study is based on a single, small dataset (n=88 participants). This is insufficient to make broad claims about state-of-the-art performance or the generalizability of the method. The reporting of results is confusing and inconsistent. Table 2 is specified as using Leave-One-Subject-Out (LOSO) validation, but the validation method for Table 1 is unstated. Furthermore, the sets of baseline models in Table 1 and Table 2 are inexplicably different.

- **Poor paper quality and structure**: The paper contains several critical structural flaws. The introduction is overly general and fails to provide any meaningful review of the specific field (self-supervised learning for EEG). Most egregiously, the conclusion introduces a new method: "Adaptive Multi-head Contrastive Learning (AMCL) strategy (Wang et al., 2024)" that I dont think is ever mentioned, implemented, or evaluated in the main body of the paper. Unless I've missed something, this suggests a very low-effort or rushed submission.

1. Mohsenvand MN, Izadi MR, Maes P. Contrastive representation learning for electroencephalogram classification. InMachine learning for health 2020 Nov 23 (pp. 238-253). PMLR.
2. Yang, C., Xiao, C., Westover, M.B. and Sun, J., 2023. Self-supervised electroencephalogram representation learning for automatic sleep staging: model development and evaluation study. JMIR AI, 2(1), p.e46769.
3. Gijsen S, Ritter K. Self-supervised Learning for Encoding Between-Subject Information in Clinical EEG. In Learning Meaningful Representations of Life (LMRL) Workshop at ICLR 2025 Mar 6.

**Questions:**

1. Could you please explain table 1 and 2? How are models in table 1 evaluated? Why are the models different between these two tables?

2. Are you able to provide (extensive/significant) additional evaluations to evaluate claims about the progress of your methods?

3. Would you be able to explain what I've misunderstood about your method/architecture and how it progress the field of EEG classification in any way?

---

### Official Review · Reviewer_oZaG · 2025-11-01

**Soundness:** 2
**Presentation:** 2
**Contribution:** 3
**Rating:** 2
**Confidence:** 4

**Summary:**

This paper proposes DGNet, a self-supervised learning (SSL) framework for dementia classification using EEG signals. The core contribution is the multi-head SimCLR architecture that is specifically designed to leverage the known neurophysiology of dementia. The key idea is to decompose the EEG signal into its five primary frequency bands.

The DGNet architecture consists of two stages:

1. Pre-training (SSL): An independent CNN encoder and projection head is used for each of the five frequency bands. The model is pre-trained on unlabeled data using a contrastive, adaptive Normalized Temperature-scaled cross-entropy (NT-Xent) loss.
2. Linear Evaluation: The pre-trained encoders for all five bands are frozen. Their output feature vectors are concatenated and fed into a simple, trainable MLP classifier for the downstream task of classifying Alzheimer's Disease (AD) vs. Cognitively Normal (CN) subjects.

**Strengths:**

1. Clear Motivation: The paper's core strength is its foundation in established neurophysiology. The architecture is explicitly designed to model the "slowing" of brain oscillations (i.e., differential changes across $\delta, \theta, \alpha, \beta, \gamma$ bands) that is a known biomarker for dementia.

**Weaknesses:**

1. Methodological Ambiguity: The paper lacks clarity on several key methodological details.

   - Training Objective: For instance, two distinct objectives are described for the pre-training stage (Section 2.2), but the connection between them is unclear, where one is using the "worst-case" negative sample while the other uses all negatives. Additionally, the regularization term mentioned in Equation 3 is only used in one of the two objectives.
   - Multi-head Architecture: The paper describes at least 2 different multi-head architectures (Section 4.1, 4.2, 4.3), namely "adaptive 5 band heads" and "Multi-head (5 heads)". It is not clear how these differ, and the attribution of each to the final results is not well explained.

2. Single Dataset Evaluation: The model is developed and validated on a single, relatively modest-sized dataset (88 subjects total). The generalizability of a model is best confirmed by testing it on an external, out-of-distribution dataset (e.g., from a different hospital, using different EEG hardware). The current experimental setup does not provide evidence of generalizability beyond the chosen dataset.

3. Untapped Data: The chosen dataset also contains data for 23 patients with Frontotemporal Dementia (FTD). The paper's experiments are limited to the binary AD vs. CN classification. Given the paper's title "Dementia Classification", it's a missed opportunity not to test the model's ability to perform the more challenging (and clinically relevant) 3-class (AD vs. FTD vs. CN) classification. This would be a valuable extension.

**Questions:**

1. Clarification on Frequency Band Extraction: This is my main point of confusion. Since this is a critical architectural detail, could you please clarify the exact procedure in Section 2.1 and Figure 2?

   - Option A: Are traditional (e.g., Butterworth) bandpass filters first applied to the raw signal to create 5 separate band-limited signals? And then each of these 5 signals is passed to its own 1D depthwise conv encoder?
   - Option B: Is the raw signal (or a single broadband-filtered signal) fed into the "frequency band extractor" module, where the "five parallel 1-dimensional depthwise convolution layers" are expected to learn band-specific features from the full signal?
   - Figure 2 seems to imply Option A, but the text is ambiguous.
   - Figure 3 on the other hand, seems to suggest Option B, since the embeddings after passing through the encoder still appear to have distinct frequency content, as "(In the) spectrogram visualization of embeddings from the encoder ..., y-axis denotes the frequency ranges corresponding to each band".

2. Clarification on Training Objective: In Section 2.2, two different training objectives are described (Equations 1 and 3). I assume that only the first objective (Equation 1) is used for pre-training the model, while the second objective (Equation 3) is only used for ablation purposes. Is that correct? If so, please clarify this in the text.

3. Use of more Data: The dataset also includes 23 subjects with Frontotemporal Dementia (FTD). Did you perform any experiments on the 3-class (AD vs. FTD vs. CN) classification task? Showing how DGNet performs on this more complex task would significantly strengthen the paper's claim of being a general "dementia classification" model. Also, evaluating your method on more datasets would significantly strengthen the evidence for the claim of generalizability.

4. Training Protocol: For the SSL pre-training stage (Fig 1a), was the model pre-trained once on the entire unlabeled dataset (i.e., data from all 88 subjects)? Or was the pre-training performed inside each LOSO fold, using only the (N-1) subjects' data for that specific fold? Although the former is a common practice in SSL, since the current evaluation relies on the same dataset, it would cause data leakage. Also, using the same set for validation and testing together with early stopping would also lead to indirect data leakage, so please clarify.

5. Clarification on Experiment Results: Section 4.1 and 4.2 describe multiple sets of experiments with different architectures. However, the distinctions between these comparisons are not very clear. For example, does the results in Table 1 use the LOSO setting? Moreover, how do the experiments in Section 4.2 (Table 2) differ from those in Section 4.1 (Table 1)? Please clarify the differences between these experiments and their results.

6. Epoch Length: Supplementary Figure 5b shows that performance increases significantly with epoch length, with 30s being the best. This makes sense, as longer epochs are needed to capture low-frequency (delta) oscillations. Did you test any epochs longer than 30s? Is it possible that 60s (a common length for sleep EEG) would be even better?

---

### Official Review · Reviewer_CaFh · 2025-11-01

**Soundness:** 1
**Presentation:** 1
**Contribution:** 1
**Rating:** 2
**Confidence:** 5

**Summary:**

The paper presents a model for dementia classification. They extract 5 bands of the EEG signal and assign an individual encoder to each band for representation learning. The SimCLR framework is used for self-supervised pre-training. The experiment is evaluated on the dataset ADFTD using Leave-One-Subject-Out (LOSO) cross-validation.

**Strengths:**

N/A

**Weaknesses:**

1) **Lack of novelty.** The overall idea lacks novelty. Extracting canonical EEG frequency bands and applying CNN-based classification is a well-established approach that has been widely explored for nearly 10 years. Similarly, the use of SimCLR for self-supervised pre-training is a classical framework that has been popular for several years. Therefore, the methodological contribution of this paper appears limited.
2) **Incorrect evaluation protocol and misunderstanding of basic concepts.** The paper demonstrates fundamental misunderstandings of the EEG domain and deep learning training protocols. In a leave-one-subject-out (LOSO) setting, one subject is held out for testing, and all remaining subjects are used for training. There is no validation set in LOSO; thus, early stopping must not be applied, as it effectively tunes on the test set, leading to severe data leakage and inflated performance. Moreover, the authors use self-supervised pre-training on all subjects, including the “left-out” subjects in the LOSO loop, which again results in information leakage. This reminds me of a funny paper several years ago called "Pretraining on the test set is all you need"[1]. Such a trick renders the reported results unreliable due to strong performance inflation.
3) **Improper preprocessing for the ADFTD dataset.** The preprocessing pipeline for ADFTD is inappropriate for an end-to-end deep-learning framework. The authors segment the EEG into 30-second windows for training, drastically reducing the number of training samples and severely disadvantaging deep models for the EEG-based dementia detection task. Such long segments are typically used only when hand-crafted features are extracted before learning (e.g., DICE-Net[2]). In contemporary end-to-end deep learning EEG models for dementia detection, segment lengths of 1-4 seconds (e.g., 200Hz sampling → 1s segments) are standard, as the ultimate goal is not for EEG sample classification but per-subject detection. For example, LaBraM fine-tuned with proper 1-second segments and without early stopping at least achieves ≥80% LOSO accuracy on ADFTD. The current setup artificially lowers baseline performance and does not provide a fair comparison.
4) **Misleading title and incomplete problem formulation.** Although the title claims dementia classification, the experiments only include Alzheimer’s disease (AD) and healthy controls (HC) from the ADFTD dataset, completely excluding frontotemporal dementia (FTD) subjects. This mismatch between title and experimental design undermines the scope and contribution of the work.


[1] Schaeffer, Rylan. "Pretraining on the test set is all you need." Arxiv.

[2] Miltiadous, Andreas. "DICE-net: a novel convolution-transformer architecture for Alzheimer detection in EEG signals." IEEE Access

**Questions:**

See weakness.

---

### Note · Authors · 2025-11-17

I have read and agree with the venue's withdrawal policy on behalf of myself and my co-authors.